# Public health concerns for food contamination in Ghana: A scoping review

Nkosi Nkosi Botha[1]*, Edward Wilson Ansah[1], Cynthia Esinam Segbedzi[1], Sarah Darkwa[2]

1 Department of Health, Physical Education and Recreation (HPER), University of Cape Coast, Cape Coast, Ghana, 2 Department of Vocational and Technical Education (VOTEC), University of Cape Coast, Cape Coast, Ghana

* saintbotha2015@gmail.com

**Data Availability Statement:** All data are included in the paper.

**Funding:** The author(s) received no specific funding for this work.

## Abstract

Nutrition is sturdily and rapidly becoming the foremost determinant of health in today's Sars-Cov-2 and climate change ravaged world. While safe food sustains life, contamination obliterates its values and could result in death and short to long term morbidity. The purpose of this scoping review is to explore food contamination in Ghana, between 2001–2022. Using Arksey and O'Malley's procedure, a systematic literature search from PubMed, JSTOR, ScienceDirect, ProQuest, Scopus, Emeralds Insight, Google Scholar, and Google was carried out. Following the inclusion criteria, 40 published and grey literature were covered in this review. The review revealed the following: Studies on food contamination involving Greater Accra, Ashanti, Central, and Eastern Regions alone account for over 50% of the total number of such studies conducted in Ghana; regulators failed in enforcing regulations, monitoring and supervision; managers failed to provide adequate infrastructure and facilities. The most common food safety risks of public health concern are: i) micro-organisms (E. coli/faecal coliforms, Staphylococcus aureus, Salmonella spp, Bacillus cereus, and Viral hepatitis); ii) drugs (Amoxicillin, Chlortetracycline, Ciprofloxacin, Danofloxacin, and Doxycycline) and; iii) chemicals (Chlorpyrifos). Salad, vegetables, sliced mango, meat pie, and snail khebab are of high public health risks. The following deductions were made from the review: Highly contaminated food results in death, short to long term morbidity, economic loss, and threatens to displace Ghana's efforts at achieving the Sustainable Development Goals (SDG) 2. Thus, Government must resource key regulatory bodies to enhance their operational capacity, regulators must foster collaboration in monitoring and supervision of food vendors, and managers of food service outlets must provide adequate facilities to engender food safety culture.

## Introduction

Poverty, climate change, conflict, rapid population growth, illiteracy, negative socio-cultural factors, anthropogenic elements individually and collectively threaten the sanctity of foods consumed globally [1, 2]. There is a strong correlation between food safety, nutrition and food

**Competing interests:** The authors have declared that no competing interests exist.

security [3], because such conditions create favourable environment for foodborne illnesses. Foodborne illnesses are infectious or toxic in origin and they are triggered by bacteria, viruses, parasites or chemical elements entering the body via contaminated foods [2–4]. Thus, fish, meat, fruits, vegetable, drinks, nuts, water, and other edibles are increasingly becoming unsafe for human consumption [3, 5]. The global food chain has become longer and convoluted amid growing consumer demand for variety of food [6].

Nutrition is fast gaining grounds as a significant determinant of public health globally amidst Covid-19 and climate change challenges [7, 8]. While safe food promotes and protects health, contamination deflates its essence resulting in loss of lives and short to long term indisposition [6]. Global estimates show that unsafe food can cause 420,000 lives loss, 125,000 under 5 deaths, 33 million healthy-life-years (DALYs), and illness of 1 in 10 individuals annually [3]. Undoubtedly, unwholesome foods have become an imminent danger to humanity, as they stretch the global healthcare systems to the brink and impede socioeconomic development, especially in the developing economies [3, 9, 10].

Annually, developing countries lose US$110 billion in productivity and medical expenses to the consumption of unwholesome foods [3]. Moreover, sub-Saharan Africa (SSA) accounts for the highest incidence of food contamination globally [11, 12]. Over 91 million SSA suffer illnesses annually from eating contaminated foods, with 137,000 deaths [11, 13]. Yet, not much is being done to guarantee food integrity and security for the population [12]. That is because the continent suffers inadequate storage capacity, poor transportation network, and poorly resourced state agencies responsible for regulating the food chain [14]. Moreover, there is significant food safety knowledge gap among most ready-to-eat food vendors in the region [15–18]. This knowledge gap can undermine food safety practices by the vendors and ultimately threaten the health and safety of the consuming public [16, 17].

Meanwhile, chronic exposure to aflatoxin and other foodborne illnesses in Africa posse economic burden, resulting in loss of productivity of $16.7 billion annually [11, 19]. The phenomenon is fast gaining attention in Africa since the establishment of the Partnership for Aflatoxin Control in Africa (PACA) in 2014 by the African Union [11]. PACA supports governments to regulate their food chain by engineering regulatory enforcement, creating awareness among small-holder farmers to control aflatoxin levels [11, 19].

In Ghana, incidence of food contamination is widespread, raising public health concerns [19, 20]. The Food and Drugs Authority (FDA), Environmental Health Management and Sanitation Units (EHS) of the Ministry of Local Government and Rural Development (MLGRD), the Ghana Tourism Authority (GTA), the Veterinary Service Department (VSD), and the Consumer Protection Agency (CPA), are the major regulatory bodies which guarantee food integrity [8, 19]. Sadly, the endless local media reports on food contamination in Ghana belie these layers of regulation [21]. Apart from its immediate impact on individuals and families, the phenomenon displaces Ghana's efforts at achieving the Sustainable Development Goals (SDG) 2 [13]. Goal 2 charged all nations to "eliminate hunger, attain food security and enhance nutrition and stimulate sustainable agriculture". Contaminated food affects the rich, poor, healthy, sick, old, young, male, female, persons with disability, educated, uneducated, and everyone. Furthermore, there are perennial outbreaks of Cholera and Typhoid in Greater Accra, and other parts of Ghana. The Ghanaian media is awashed with reports on food contamination, indicating the public health and safety relevance of the issue. Thus, what risk factors account for this phenomenon? What is the prevalence of food contamination in Ghana? Which ready-to-eat foods pose the highest public health and safety risk in Ghanaians? What preventive measures could help curb this canker?

### Rationale

Several studies reported incidence of food contamination, microbial examination of foods, and food poisoning in Ghana. Yet, the problem persists with increasing prevalence and threat to public health and safety. Therefore, there is need to explore existing evidence, highlight the areas of public health relevance and make recommendations for consideration.

### Objective

The objective of this scoping review is to synthesis existing evidence on the phenomenon of food contamination in Ghana, with the view to contributing to knowledge on the subject.

## Methods

We relied on published and grey literature to examine the prevalence and factors affecting food contamination in Ghana. Leveraging on Arksey and O'Malley's [22] procedure, we synthesized and analysed wide spectrum of useful published and grey literature from 2001 to 2022. The procedure includes: i) exploration and crafting of review objectives; ii) exploring vital literature; iii) sorting of relevant literature; iv) data mining; v) summary of data and synthesis of results, and vi) consultation [23]. Therefore, we stated four research questions: i) What is the prevalence and regional distribution of food contamination in Ghana? ii) What are the food-safety-related public health risks of ready-to-eat foods in Ghana? iii) Which foods are of high public health risk in Ghana, and What are the microbial qualities of ready-to-eat foods in Ghana? iv) What are the preventive measures against food contamination in Ghana?

### Search strategy applied

Consistent with the Preferred Reporting Items for Systematic Reviews and Meta-Analyses (PRISMA) strategy, we searched for useful published and grey literature for this review [24, 25] (see Fig 1).

The data search went through two levels. At level one, we applied words like "food" OR "contamination" OR "risk" OR "Ghana", which is intended to gauge the volume of data available on the subject. The results from the initial search included: PubMed (235), JSTOR (1,466), ScienceDirect (59,120), ProQuest (44,177), Scopus (59,120), Emeralds Insight (227), Google scholar (91,200), Google search (3,650,000). With additional words (see Table 1) added, a second level search was conducted and returned the following: PubMed (86), JSTOR (94), ScienceDirect (245), ProQuest (211), Scopus (112), Emeralds Insight (82), Google scholar (127), Google search (405). In all, 1,362 articles were located. The search exercise spanned April 21st, 2022, to June 19th, 2022.

In keeping with the inclusion norms, 40 published articles were included in this review (see Table 1). All four authors, "EWA", "CES", "SD", and "NNB" individually extracted data from published articles that agreed with the inclusion norms for this scoping review. Meanwhile, "SD" and "EWA" fixed inconsistencies that emerged during data extraction. We categorized the phenomenon of food contamination in Ghana into five main themes, i.e.; i) prevalence and regional distribution, ii) causes, iii) microbial and other test results, iv) high risk foods, and v) prevention (see Table 2).

## Results

The study took a retrospective glance at articles –40 (23 published and 17 grey), covering 2001 –June, 2022, and found two main designs, quantitative, 28 and mixed method, 12 studies.

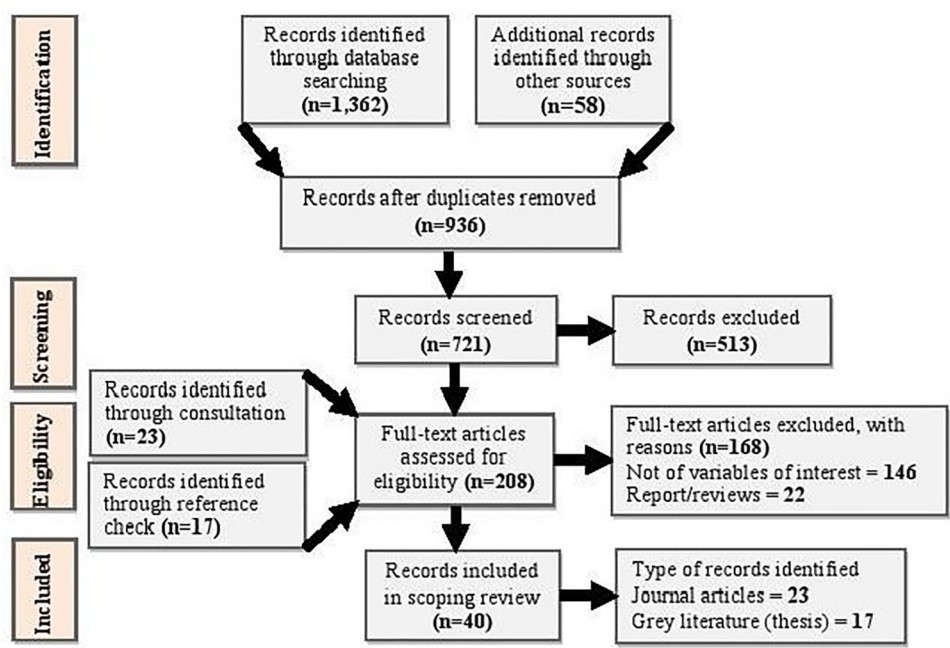

**Fig 1. PRISMA flow diagram.**

**Table 1. Search strategy.**

| Search Strategy Item | Search Strategy |
|---|---|
| Search engines | PubMed, JSTOR, ScienceDirect, ProQuest, Scopus, Emeralds Insight, Google Scholar, Google Search. |
| Language filter | English |
| Time filter | 2001—June, 2022 |
| Spatial filter | Ghana |
| | "food" OR "nutrition" OR "meals" OR "diet" OR "breakfast" OR "lunch" OR "dinner" OR "supper"<br>"contamination" OR "pollution" OR "adulteration" OR "infection" OR "doctored" OR "poisoned"<br>"microbes" OR "bacteria" OR "germs" OR "virus" OR "fungi" OR "bugs"<br>"quality" OR "classic" OR "excellent" OR "superior"<br>"risk" OR "danger" OR "hazard" OR "threat" OR "menace"<br>"preventive" OR "defensive" OR "deterrent" OR "protective" OR "intervention" OR "solution"<br>"measures" OR "methods" OR "procedures" OR "steps" OR "arrangements" OR "techniques"<br>"Ghana" |
| Inclusion criteria | 1. Published articles on Ghana; 2. grey literature on Ghana; 3. articles on food safety in Ghana covering 2001 –June, 2022; and 4. articles must provide details on methodology, sample/population, study area, causes of food contamination, microbial isolation, high risk foods, prevention of food contamination, and conclusion. |
| Exclusion criteria | 1. Published and grey articles on food safety articles before 2001; 2. articles on food safety in other countries except Ghana; 3. articles lacking details on author(s), methodology, population, study area, causes, microbial isolation, high risk foods, prevention of food contamination, and conclusion, 4. review articles. |

**Table 2. Extracted data.**

| Author | Purpose | Methods | Prevalence/ Population/ Sample | Causes of food contamination | Microbial/ other test results | High risk foods | Prevention | Conclusion |
|---|---|---|---|---|---|---|---|---|
| Abakari et al. (2018) | Microbiological quality of pre-cut vegetable salads | Quantitative | Four zones/strata (1-Timber market, Transport Yard and Nyohini; 2- Zobgeli, Aboabo and Sabonjida; 3-Changli, Taxi Rank, Bus Stop and Tishigu; and 4-Moshie Zongo, Gumbihini and Parks and Gardens), Tamale Metroplis, Northern Region. Salads (cabbage, lettuce, onions and tomatoes mixtures). Samples of salad vegetables (30) (strata one—7, strata two—9, strata three—8, strata four—6) | Improper food handling, unhygienic food preparation and processing, transmission and source contamination of the vegetables | *Escherichia coli*, *Bacillus cereus*, *Salmonella spp.*, and *Shigella spp* | Salad | Enforcement of regulation and laws | Salads sold were unwholesome for human consumption and could be deleterious to the health of consumers |
| Addo-Tham et al. (2020) | Knowledge about food safety and food handling practices | Mixed | 340 participants, Ejisu-Juaben Municipality, Ashanti Region | Not stated | N/A | N/A | Strengthen training for food vendors through collaboration between municipal assemblies and other agencies | Although the majority of the vendors lacked tertiary education, their food safety knowledge is good and should be respected |
| Aduah et al. (2021) | Vendor and consumer perception, knowledge, and practices of meat safety regarding ready-to-eat (RTE) meat and how this affected the prevalence and antibiotic susceptibility of *Salmonella enterica* in RTE meats | Quantitative | Ready-to-eat meat vendors (300) and consumers (382). Beef (50), chevon (50), chicken (50), guinea fowl (50), mutton (50), and pork (50), Bolgatanga, Upper East Region | Cross-contamination from faulty handling | *Salmonella enterica* | Guinea fowl and beef | Continuous surveillance of *Salmonella enterica* incidence in meat products | The good knowledge and practice of meat safety demonstrated by the respondents corroborated the negligible prevalence of *Salmonella*, reiterating the importance of vendor meat safety knowledge. However, the presence of resistant *Salmonella enterica* in some of the meat samples, albeit in a very low prevalence, warrants stricter sanitary measures and greater meat safety awareness in the general population to prevent meat-borne infections and potential transmission of drug-resistant bacteria to humans |
| Aglidza, E. M. (2019) | Factors that influence the choice of location for street food vending, safety practices of handling and serving of food and sanitary conditions under which street food vendors operate | Quantitative | 85 informal street food venders, Sekondi, Western Region | Inadequate time and temperature control of food, as well as cross-contamination. | N/A | N/A | Health authorities should enforce laws on street food vending to ensure that street food vendors located their trade in sanitary environment to prevent possible transfer of infections | Most significant factors influencing the choice of location of food vendors is proximity to customers, and therefore vendors did not consider sanitary conditions around their vending points. Thus, the safety of food vendors' food handling practices was compromised |

*(Continued)*

**Table 2.** (Continued)

| Author | Purpose | Methods | Prevalence/ Population/ Sample | Causes of food contamination | Microbial/ other test results | High risk foods | Prevention | Conclusion |
|---|---|---|---|---|---|---|---|---|
| Agyarko, D. A. (2021) | Microbial and chemical contaminants in "mashed kenkey" | Quantitative | 10 milling operators and 5 "mashed kenkey" vendors. Samples tested (water from milling plants, unmilled kenkey, milled kenkey, and final processed milled kenkey), New Juaben, Eastern Region | Milling plants suspected to be polluted with microbiological and chemical contaminants | Coliforms, *E. coli*, *Staphylococcus*, yeast and moulds | Mashed "kenkey" | Safe personal and food hygiene practices, regular cleaning of milling plant, clean water, enforcement of regulation, culprits punished | Aerobic mesophylls found but at safe levels. Faecal and other common contaminants within acceptable levels |
| Ahmed (2018) | Factors that contribute to food and water borne illnesses among households | Quantitative | Householders (230), Ashaiman Municipality, Greater Accra Region. | Poor food storage, poor environmental and food safety. | N/A | N/A | Public education on food and water safety, and effective waste management | Socioeconomic factors such as employment status and housing /environmental factors such as storage of fresh/ left-over food on shelf were associated positively with reported food and water borne illnesses among the sampled residents. Most importantly, the availability of water and sanitation coupled with proper food and water safety can prevent the occurrence of food and water borne illnesses among households |
| Akabanda et al. (2017) | Food safety knowledge, attitudes, and practices | Quantitative | 235 institutional food-handlers from 29 institutions, Upper East Region | Poor food safety hygiene practices | N/A | N/A | Continuous food safety education and motivation for food handlers of various demographic backgrounds with special attention on those with lower levels of education | Institutional food-handlers had satisfactory knowledge in the areas of food safety, personal hygiene, cleaning and sanitation procedures. However, this did not translate into strict food hygiene practices. |
| Akansale (2019) | Knowledge of farmers in antibiotic usage and the presence of antibiotic residues in chevon (goat meat) and beef | Quantitative | 150 farmers interviewed; 36 samples (18 beef and 18 chevon). Sunyani, Now Bono Region | Unhygienic conditions, and treating animals with antibiotics. | Amoxycilin, chlorotetracycline, ciprofloxacin, danofloxacin, doxycycline, norfloxacin, oxytetracycline, sulfadiazine and tylosine | N/A | N/A | Farmers had limited knowledge in antibiotic usage and some antibiotic residues were present in chevon and beef sold. Concentrations of the various antibiotics residues were below the maximum residue limit and can be considered safe for human consumption |
| Akonor et al. (2013) | Domestic food handlers' knowledge on food safety | Quantitative | 608 domestic food handlers, from 180 homes, Greater Accra | Not stated | N/A | N/A | Food safety education | Demographic factors that significantly influence food safety knowledge were education and age. Respondents were well-informed about food safety, general and personal hygiene |

*(Continued)*

**Table 2.** (Continued)

| Author | Purpose | Methods | Prevalence/ Population/ Sample | Causes of food contamination | Microbial/ other test results | High risk foods | Prevention | Conclusion |
|---|---|---|---|---|---|---|---|---|
| Amaami et al. (2017) | Factors associated with poor compliance of street food vendors to safety measures, with emphasis on the WHO's five keys to safer food policy | Quantitative | 150 respondents (140 food vendors and 10 officials of the Environmental Health and Sanitation Agency), Techiman Municipality, now Bono East Region | Inadequate funds, poor regulation, and lack of logistics and resources by regulatory bodies | N/A | N/A | Current regulations in the municipality regarding general food hygiene practices should be reviewed and strengthened to focus on a risk-based approach | High awareness of food safety among food vendors in the Techiman municipality. Yet food hygiene and handling practices are poor |
| Amedewonu (2020) | Food hygiene practices and level of awareness about food-borne diseases | Mixed | 115 food vendors, Caoltar Township, Ayensuano District, Eastern Region. | Not stated | N/A | N/A | Registration and certification, leftover foods should be avoided, regular education and training, good personal and food hygiene practices, monitoring and supervision | Food vendors did not practice positive food handling resulting in food-borne diseases. This is due to irregular monitoring and supervision, poor training, and gross disobedience by food vendors. Food vendors practiced poor food hygiene and insanitary practices. Majority of them were medically screened and certified |
| Ameme et al. (2016) | Determine the magnitude and source foodborne diseases and implement control and preventive measures | Retrospective cohort study | Adeiso, Upper West Akim District, Eastern Region. Community members and patrons of food joints | Not stated | N/A | Waakye and shitor | Effective FBD surveillance system complemented by a strong laboratory capacity | A point source foodborne disease outbreak linked to probable contaminated "waakye" and or "shitor" occurred. Missed opportunities for definitive diagnosis highlighted the need for strengthening local response capacity |
| Annor et al. (2011) | Food hygiene knowledge, attitudes and practices, and the microbiological load of foods | Quantitative | 42 food handlers from six hotels in Accra. Rice (plain white rice, jollof rice, and fried rice), meat or poultry (boiled, and poultry was either fried or grilled), tomato stew/sauce, and Shito | Poor food hygiene | Coliforms (*E. coli*) and *Staphylococcus aureus* in food from four out of the five hotels | N/A | Strict food hygiene practices, managers should develop strategies to motivate their food handlers to practice food hygiene | Food hygiene knowledge and attitudes were satisfactory; however, its practice was challenging. Gender, age and educational level of respondents did not influence their food hygiene practices. Microbial counts of all food samples were generally high. Food hygiene knowledge and attitudes of the food handlers did not result in efficient food hygiene practices. Consumers were likely to be exposed to contaminated food |

(*Continued*)

**Table 2.** (Continued)

| Author | Purpose | Methods | Prevalence/ Population/ Sample | Causes of food contamination | Microbial/ other test results | High risk foods | Prevention | Conclusion |
|---|---|---|---|---|---|---|---|---|
| Aovare (2017) | Food vending and hygiene practices | Mixed with microbial test | Total of 150 stationary street food vendors and restaurants, i.e. 15 from 10 communities (Tindonsobulugu, Yarigabisi, Zuarungu Dachio, Gambibigo-Azuabisi, Kumbosigo, Sherigu Dorungu-Agobgabis, Zaree, and Atulbabisi). 66 food samples (rice balls and groundnut soup; Tuozaafe [TZ] and vegetable soup; jollof rice; kenkey and tomatoe sauce; and porridge [kokol] Bolgatanga municipality. | Poor dish washing practices, recycling of washing water, and the multiple use and reuse of napkins | *Salmonella typhi, Escherichia coli, Staphylococcus aureus* and fecal coliforms | Not stated | Frequent media and personal education, safe food handling, and also consumers on identifying safe food, proper personal and food hygiene, and enforcement of regulations | The most significant factor influencing the choice of location of food vendors is proximity to customers. The physical state of the environment was satisfactorily healthy. Food safety handling practices were compromised. Institutional set-up was not effectively promoting conformance to food vending controls. Most of the foods were contaminated above acceptable levels of consumption |
| Appietu et al (2020) | Food safety knowledge, the microbiological quality of school meals, as well as the barriers to food safety practices. | Quantitative | 97 food handlers (5 boarding senior high schools) Ho, Volta Region. 60 food samples (4 cooked foods) | Inadequate provision of equipment and irregular water supply | *Escherichia coli, Staphylococcus aureus, Pseudomonas spp.,* and *Bacillus cereus* | Not stated | Administrators of SHSs must design and implement food hygiene training and sanitation programmes for food handlers in schools. also, more stringent supervision during food preparation processes and routine microbial analysis of cooked foods | Pathogenic bacteria can exist in cooked foods, even though they may physically appear to be quite wholesome |
| Ayensu (2020) | Major hygiene practices and other related factors | Quantitative | 370 uncooked vegetable and fruit sellers, Ashaiman Central Market, Ashaiman Municipality, Greater Accra Region | Poor food hygiene, irregular water supply. | N/A | Not stated | Routine food inspection, continuous supervision, provision of waste bins, regular water supply, and sustained food hygiene education | Vegetable and fruit vendors practiced good hygiene with nearly half of them having adequate knowledge of good hygiene practices. Gender, type of uncooked/ raw foodstuff sold, mode of preservation of vegetables and fruits and water supply at selling points were factors associated with hygiene practices among the vegetable and fruit vendors. Generally, sanitation of the place where most vendors sold their uncooked food was clean and neat, though this could still be improved |
| Ayimpokaapegyine (2016) | Food hygiene practices of food service establishments (FSE) in the hospitality industry | Quantitative | FSEs (97), localities (11) (namely, Dansoman, East Legon, Osu, Odawna, Ayi Mensa, Adenta, Dome, Madina, Taifa, Haasto, and Ashongman), Greater Accra Region | Lack of basic infrastructural needs and utilities, inadequate personal, food, and environmental hygiene practices, inadequate training | N/A | N/A | Increased and sustained provision of tailor-made sensitization and training programmes | Inadequate food safety practices with respect to food hygiene and storage practices. Though food handlers had good food safety knowledge, their practices were inadequate |

*(Continued)*

**Table 2.** (Continued)

| Author | Purpose | Methods | Prevalence/ Population/ Sample | Causes of food contamination | Microbial/ other test results | High risk foods | Prevention | Conclusion |
|---|---|---|---|---|---|---|---|---|
| Bansah (2018) | Food safety and hygienic practices | Mixed | Accra, Greater Accra Region; Takoradi, Western Region; and Senchi, Asuogyaman District, Eastern region Sample of 300 (150 street food vendors and 150 consumers). 50 food vendors and 50 consumers each from each study area, Makola-Accra, Takoradi market circle and Senchi road side. 116 food samples from 150 street food vendors, 26 each per season for the wet and dry seasons. Foods include Jollof rice, Meat Pies/ Spring rolls, Spinach/ Okro soups, Plain/Salads with cream, tomato-pepper sauce/stew/ black pepper sauce (shitor), sliced fruits like water melon, pineapple, pawpaw, and mango, mixed fruits of pineapple, water melon, mango and banana, "indomie", fried yam, roasted plantain, and snail "khebab" | Poor personal/food hygiene and poor temperature control of food due to frequent power outages ('Dumsor') | Coliform and *Salmonella sp.* in food/ water | Sliced mango, meat pie, snail khebab, sliced mango, fried yam, sliced water melon, jollof, roast plantain and plain salad most contaminated | National Sanitation Day to promote sanitation and good health. regulatory bodies such as the Environmental Protection Authority, Food and Drugs Authority and the Metropolitan/ Municipal/District Assemblies (MMDAs) should be more aggressive in enforcing food safety standards | Age and education important factors for food safety perception while education and training on food safety were factors influencing attitude on good hygiene practices. Majority of respondents had limited knowledge on microorganisms and their effect on human health |
| Darko et al. (2017) | Food safety standards practiced and mould contaminants | Microbial testing | Forty food samples from five hotels in Kumasi, Ashanti Region | Poor food hygiene, inadequate processing, unwholesome food ingredients, and poor kitchen sanitation | *Eurotium herbariorum, Aureobasidium pullulans, Alternaria alternate, Botrytis cineria and Fusarium oxysporu* | Salads and vegetable dishes | Hotel inspections should include microbial test on foods | It was observed that foods tested were above the acceptable levels and could be sources of food borne pathogens. It is recommended that hotel inspections should include microbial test on foods |
| Darko et al. (2015). | Knowledge of kitchen staff on food safety as well as kitchen hygiene | Quantitative | 39 hotels (three stars and budgets), kitchen staff, Ashanti Region | Not stated | N/A | Not stated | Regular evaluation of kitchen to ensure good hygiene practices | People within the high socioeconomic group as well as foreigners patronize prepared food from the hotels with the assumption that foods sold from these hotels are prepared under good hygienic conditions. Also, that food handlers will practice food safety. Most hotel kitchen staff had adequate knowledge of food safety and kitchen hygiene |

(*Continued*)

**Table 2.** (Continued)

| Author | Purpose | Methods | Prevalence/ Population/ Sample | Causes of food contamination | Microbial/ other test results | High risk foods | Prevention | Conclusion |
|---|---|---|---|---|---|---|---|---|
| Ekow (2018). | Food safety management systems and hygiene practices | Quantitative | 100 respondents (senior high schools–Accra Girls, St. Marys girls, Achimota, Accra Academy, O'reilly, St. Marys Girls, and Presby Boys-Legon). Students, Matrons, cooks, dining hall masters, store keepers, and pantry servers and team leaders interviewed | Inadequate food service facilities and equipment; weak monitoring systems by AMA | N/A | N/A | Schools be mandated to adopt and implement food safety management programs to safeguard the health and wellness of students | Generally, the schools have poor food safety systems and hygiene practices. There is potential for outbreak of foodborne incidents or food safety incidents. Government and Academia to collaborate in implementing food safety management systems in second cycle institutions |
| Feglo et al. (2012) | Bacterial isolates and total counts of bacterial species responsible for contamination of street vendored food | Quantitative with bacterial isolation | 60 food samples [ice-kenkey (15), cocoa drink (15), fufu (5), ready-to-eat red pepper (5), salad (10), and macaroni (10)]. Kejetia, Asafo, Race course, and Bantama bus terminals, Food vendors in Kumasi, Ashanti Region | Poor sanitary conditions, food hygiene and sanitation | *Staphylococcus aureus*, and *Escherichia coli* | Macaroni, salad, fufu and cocoa drink | Water for cocoa drink and ice-kenkey should be treated. Vegetables should be washed with treated water. Food hygiene education | Most ready-to-eat foods in Kumasi were contaminated with enteric bacteria and other potential food poisoning organisms with bacterial counts higher than the acceptable levels. Street foods therefore pose a health threat to the patrons and efforts to reduce level of contamination recommended |
| Ghartey (2019) | Street-vendor food risk factors and regulation/ enforcement practices | Mixed | 413 [266 and 147 for Komenda Edina Eguafo Abirim Municipality (KEEA) and Ajumako Enyam Essian District (AEE), respectively]. Central Region | Low level of formal education, poor sanitary environmental conditions, display of foods in open air, poor hand hygiene, open defecation, poor waste disposal, and lack of basic sanitary facilities | N/A | N/A | Enforcement of food safety regulations, logistical support for regulatory authorities, provision of sanitary facilities | Overwhelming proportion of street food vendors were not formally educated. One-fifth operate under insanitary environmental conditions, and three-quarters display foods in open air. Enforcement of food safety regulations was weak, and consumers placed more premium on socioeconomic attributes of street food rather than on safety |
| Kasu et al. (2022). | Determine the magnitude of food poisoning and the causative agent, and instituted control measures | Quantitative | Food samples from homes of affected victims (22) and blood sample from index case, residents of Akakpokofe, South Tongu District, Volta Region | Unintentional and accidental ingestion of food contaminated by Chlorpyrifos containing pesticides and other organophosphates | Chlorpyrifos found | Not stated | Health education of farmers and community members on appropriate use of pesticides, enforcement of laws and policies on appropriate use of pesticides | Chlorpyrifos (used for agricultural purposes) found and contaminated food eaten by two households. However, how the food got contaminated was undetermined during the investigation. Patients delayed three days before seeking health care contributing to high mortality |

*(Continued)*

**Table 2.** (Continued)

| Author | Purpose | Methods | Prevalence/ Population/ Sample | Causes of food contamination | Microbial/ other test results | High risk foods | Prevention | Conclusion |
|---|---|---|---|---|---|---|---|---|
| Konlan (2019) | Availability of nutrition standards and to assess the nature of foods sold in canteens | Mixed | Food outlets (98), canteens—79.6%, were table top foods—20.4%. Key informants (9), Physical, Development and Municipal Services Directorate (PDMSD) - 3, and University Legal Council -1, University of Ghana main campus Greater Accra Region | Lack of toilet facilities | N/A | N/A | Regulation of canteens, toilet facilities at all canteens, improved food safety and hygiene, medical screening for vendors, additional dieticians | The University of Ghana canteens sell energy dense meals and substantial consumption of which increases one's risk to NCDs. The majority of the food outlets were involved in the sale of sugar sweetened beverages, fried foods, refined carbohydrates compared to the sale of fruits and vegetables and whole grains. The overall food hygiene and safety practices of the canteens were fair |
| Kortei et al. (2020) | Microbiological quality of some ready-to-eat mixed vegetable salad, ingredients, and water samples | Quantitative | 30 food samples (mixed vegetable salads, foods and water), Accra | Poor hygiene practices | *E-coli, Bacillus cereus, Clostridium perfringens, Staphylococcus aureus, Enterobacteriaceae* | Not stated | Good hygienic practices | Bacterial loads on mixed vegetable salads, ingredients and water samples used and served were within safe limits and therefore good for human consumption |
| MacArthur (2007) | Compliance with food safety measures by traditional caterers | Mixed | 254 subjects (100 caterers, 150 clienteles [90 patrons, 30 workers and 30 students of UCC], and 4 regulatory agencies), Cape Coast, Central Region | Poor food handling, lack of food safety information, poor sanitary conditions, weak monitoring, lack of coordination and collaboration among regulatory bodies | *Salmonella typhi, Aspergillus flovlis, Fusarium spp, Penicillium spp, Rhizopus spp, Fusarium* and Coliform bacteria | Not stated | Enforcement of regulation, recalcitrant vendors should be sanctioned, good food handling practices, training, | Traditional caterers had low educational backgrounds and are ignorant about food safety information. However, there was no indication that any of the background characteristics used significantly influence compliance with food safety measures. Lack of co-ordination among regulatory bodies contributed to duplication of tasks and subsequent negligence of duty |
| Manko (2018) | Hygienic practices of food vendors and its effects on consumer food safety | Mixed | Food vendors, 31(Night market [16] and Bush canteen [15]). Food outlets (10), 5 each from the Night market and Bush canteen, Grounds and Environmental Health Services Officers (2), and student consumers (300), University of Ghana, Legon, Greater Accra Region | Inadequate training, few health services officers to embark on regular inspections, weak adherence to proper hygiene, and refusal to apply knowledge about hygienic practices | N/A | N/A | Frequent and adequate training, regular inspection of food vendors, adherence to good personal and food hygiene, culprits to be sanctioned | Food vendors do not engage in safe food practices even though they are knowledgeable about safe food practices. Most of the students suffer from food poisoning. Though training needs are organized for food vendors, it is inadequate to equip them with knowledge and skills needed |

(*Continued*)

**Table 2.** (Continued)

| Author | Purpose | Methods | Prevalence/ Population/ Sample | Causes of food contamination | Microbial/ other test results | High risk foods | Prevention | Conclusion |
|---|---|---|---|---|---|---|---|---|
| Mensah et al. (2002). | Microbial quality of foods on streets and factors predisposing to their contamination | Quantitative | Street foods vendors (120), food samples (511), Nima-Kotobabi-Pig Farm-Accra New Town sub-area, Greater Accra Region | Defective food, personal, and environmental hygiene, | Mesophilic bacteria, *Bacillus cereus*, *Staphylococcus aureus*, *Enterobacteriaceae* | Salads, macaroni, fufu, omo-tuo, red pepper, akple, rice, and waakye | Stricter implementation of food sanitation code, licensing of street food vendors, monitoring, enhanced personal, food, and environmental hygiene, consumer pressure | Street foods can be sources of enteropathogenesis. Vendors should be trained in food hygiene. Special attention should be given to the causes of diarrhoea, the transmission of diarrhoeal pathogens, the handling of equipment and cooked food, hand-washing practices and environmental hygiene |
| Monney et al. (2014). | Compliance with eight food hygiene and safety principles and probe existing institutional and legislative frameworks for regulating the activities | Quantitative | Respondents (200), Bibiani of Bibiani-Anhwiaso-Bekwai District, Now Western North Region, and Dormaa-Ahenkro of Dormaa Municipal, Now Bono Region | Weak adherence to protective clothing and head covering use. | N/A | N/A | Capacity building and harmonisation of institutional roles and legislations, and intensive public education on food hygiene and safety principles | Generally, compliance was marginally good with internationally recommended guidelines for food hygiene and safety. Compliance at Dormaa-Ahenkro was relatively better and higher than Bibiani. However, use of protective clothing and head covering were poorly complied with. Weak institutional capacities; logistical constraints; overlapping and duplicated institutional responsibilities; inconsistent local bye-laws were reported |
| Mwini et al. (2019) | Sources of microbial contamination in the production of *Wagashi-Cheese* | Mixed | All cow milk and *wagashi* producers (80) (milk producers –40 and *wagashi* –40). Samples of milk and *wagashi* (60). Sissala East District, Upper West Region. | Dirty cows, unclean containers for receiving milk, and improper handling of milk while transporting to Wagashi-Cheese centres | Coliform, fecal coliform, *Escherichia coli*, mesophilic (PCA), yeast and mould | Not stated | Training, good personal and food hygiene. | Milk and *Wagashi-Cheese* producers do not practice optimal personal, food and environmental hygiene |
| Odonkor et al. (2020) | Food safety knowledge and practices | Quantitative | 306 food handlers in hotels, restaurants, fast food joints, and chop bars, Central Region | Lack of training and education, and annual medical screening | Diarrhoea, cholera, and typhoid were self-reported | N/A | Education and awareness about food safety and hygiene. Further, effective monitoring of vendors, and strict law enforcement | There is moderate level of awareness of food safety practices among food handlers. Also, most of them from restaurants had knowledge about food safety principles. There is significant association between sociodemographic characteristics and food safety practices |
| Osei-Tutu et al. (2019) | Trends and patterns of foodborne diseases | Quantitative | Patients who visited Ridge hospital during 2009–2013. 46,432 patient records (2009–2013) reviewed, Greater Accra Region | Poor food hygiene, drug resistance, and poor health care infrastructure. | Cholera, typhoid fever, dysentery [shigellosis], and viral hepatitis. | N/A | N/A | The commonly reported foodborne diseases at the Ridge Hospital were: typhoid fever, dysentery, cholera and viral hepatitis. These diseases were found to be very seasonal with peaks at the onset of the rainy season |

*(Continued)*

**Table 2.** (Continued)

| Author | Purpose | Methods | Prevalence/ Population/ Sample | Causes of food contamination | Microbial/ other test results | High risk foods | Prevention | Conclusion |
|---|---|---|---|---|---|---|---|---|
| Ovai et al. (2019) | Food safety knowledge, attitudes and practices | Quantitative | Informal live bird traders (132) from 12 markets, Accra | Not stated | N/A | Not stated | Education on food safety practices and personal hygiene | The traders' attitude towards food safety was generally positive. Significant gaps were observed in relation to ignorance of sources of contamination during primary poultry processing operations, infrequency of hand washing before and during poultry processing, infrequent washing of food contact surfaces and carcass showering after evisceration, and complete disregard for carcass chilling |
| Owusu (2015) | Microbiological quality and safety of cooked foods and their safety. Also, level of knowledge of foodborne diseases and food hygiene | Mixed | Respondents 428 (300 students and 128 food vendors). Schools (3) from 4 urban educational circuits (Adukrom, Akropong, Larteh and Mampong). Food samples (288) ('waakye', 'shito', red pepper sauce, macaroni, fried fish, vegetable salad, sausage, groundnut soup, 'banku', iced 'kenkey', 'kenkey' and 'fufu') from each school. Akuapem North Municipal Assembly, Eastern Region | Lack of knowledge about foodborne diseases, poor personal and food hygiene, and use of untreated water | *E. coli*, coliform, *B. cereus*, *S. aureus*, yeast, and *Salmonella spp* | Fufu, pepper sauce and vegetable salad Macaroni, red pepper, sausage, and fried fish | Medical screening, education on personal and food hygiene, certification, monitoring and supervision, and regular quality checks of school canteens by regulatory authorities and School Health Education Programme (SHEP) coordinators | Food vendors lack adequate knowledge on food hygiene and foodborne disease. However, students had adequate knowledge on same. The microbial contamination of most of the cooked foods was high in the afternoons compared to the morning. All circuits recorded levels of contamination higher than the acceptable values. Food samples such as waakye, iced kenkey, banku, and kenkey recorded relatively acceptable microbial contamination both in the morning and afternoon |
| Quartey (2018) | Knowledge on food safety, practices, and the relationship between knowledge and practice | Quantitative | 200 meat vendors in Turaku slaughter slab and Madina market, Kpone-Katamanso District Assembly and La-Nkwantanang Madina Municipality, Greater Accra Region | Inadequate personal hygiene practices | N/A | Not stated | Food safety training, good personal, meat, and environmental hygiene, inspection, and culprits sanctioned | Though knowledge of foodborne pathogens and high-risk groups were unsatisfactory, raw meat handlers affirmed some responses. Meat handlers had poor food safety practices. About half of them sometimes handled meat with sores on their hand and used only one towel to clean at the work place |

(*Continued*)

**Table 2.** (Continued)

| Author | Purpose | Methods | Prevalence/ Population/ Sample | Causes of food contamination | Microbial/ other test results | High risk foods | Prevention | Conclusion |
|--------|---------|---------|-------------------------------|------------------------------|-------------------------------|-----------------|------------|------------|
| Segbedzi et al. (2022) | Food safety knowledge, attitudes and practices of chopbar workers, and the relationship between knowledge and practices on food safety | Quantitative | 147 chopbar workers, Komenda Edina Eguafo Abirem (KEEA) Municipality, Central Region | Inadequate personal and food hygiene practices, inadequate training, and low level of education | N/A | N/A | Formal training, food safety and personal hygiene practices | Chopbar workers have low levels of knowledge on food safety. Most of them had no formal education and this impacted negatively on their level of knowledge. However, majority of them had a positive attitude towards food safety practices, as many of them demonstrated a moderate level of food safety practice |
| Seidu (2020) | Food safety knowledge and practices | Quantitative | Food handlers (214) and restaurants (23), Tamale Metropolis, Northern Region | Poor work schedules, inadequate equipment and supplies, inadequate knowledge and skills, no motivation, inadequate personal, food, and environmental hygiene, and lack of enforcement of rules and regulations | N/A | N/A | Monitoring and supervision, health education, training, appropriate and adequate supply of resources, enhanced personal and food hygiene practices and capacity building for facility managers | Food handlers have high food safety knowledge compared to practice. They were limited in their sources of information, and food safety practices inadequate |
| Tuglo et al. (2021) | Knowledge, attitude, and hygiene practices of food safety | Quantitative | Food handlers (407), North Dayi District, Volta Region | Weak food hygiene practices | N/A | N/A | Food safety and hygiene training | Food handlers had good knowledge on food safety. Registering as a food handler was significantly associated with good knowledge and hygiene practices of food safety |
| Wuliyeng (2013) | Hygiene and safety practices and factors that inform these practices | Mixed | Street food vendors (72), Nsawam, Adoagyiri, Eastern Region | Weak hygiene, safety, food handling and cooking skills/ practices, lack of formal knowledge/ training on food safety, poor infrastructural and social amenities such as pipe borne water, modern toilet facilities, waste collection services, electricity, poor monitoring and controls of vendors, and ineffective enforcement of rules and regulations | N/A | N/A | District assemblies to establish food vending sites with good social amenities and infrastructure, capacity building and logistics for departments and units tasked with regulating the street food sector, provide capacity building for vendors | Weak hygiene practices exist among food vendors. Most do not wash their hands during food preparation and sale. Vendors also use their bare hands to serve customers while using those same hands to receive money |

Specifically, articles included cover 2002 to June, 2022, i.e.; 2020 and 2019–7 each, 2018–6, 2017–4, 2021–3, 2022, 2016, 2015, and 2013–2 each, 2014, 2012, 2011, 2007, and 2002–1 each.

## Prevalence and regional distribution of studies on food contamination

We found that there were far more studies on food contamination in Greater Accra than in any other region in Ghana. Thus, Greater Accra [26–39], Ashanti [40–44] and Eastern [31, 45–48], Central, [49–52], Bono [20, 53, 54] and Volta [55–57], Northern [8, 58], Upper East [59, 60] Upper West [61, 62], and Western [31, 63], and Western North [20]. Out of the 11 regions reported in this review, four (Ashanti, Central, Eastern, and Greater Accra) alone account for 70% of the incidence of food contamination in Ghana.

## Food safety-related public health risks in Ghana

We found a number of food safety-related risks of public health concern in Ghana. First, weak enforcement of regulations –3 [8, 48, 54], inadequate monitoring and supervision –2 [32, 50], and logistical limitations and ineffective collaboration –2 [50, 54] by the regulatory authorities. Second, lack of infrastructure and facilities –12 [8, 22, 29–33, 45, 47–49, 55], inadequate training –3 [30, 35, 51], inadequate monitoring and supervision –2 [48, 50], hiring of inadequate personnel and others –1 [35] by the managers. Third, poor personal and food safety hygiene practices –26 [8, 20, 26, 28–31, 34–37, 39, 41–44, 47, 48, 50, 52, 57–60, 62, 63], poor environmental sanitation –12 [8, 26, 30, 36, 42, 44, 45, 49, 50, 53, 58, 62], inadequate knowledge and skills –5 [8, 35, 47, 48, 52], inadequate time and temperature controls –4 [26, 31, 59, 63], insufficient formal education and training –3 [49, 51, 52], and lack of medical screening –1 [51] by food handlers. Fourth, consumers' poor personal/food hygiene and safety practices –5 [28, 30, 39, 42, 60], inadequate time and temperature controls –1 [31], and others –2 [49, 56]. These factors provide conducive environment for food contamination.

## Microbial quality and foods with high public health risks

The review further identified food contaminants of public health concern in ready-to-eat foods. First is bacteria: *E. coli*/faecal coliforms –11 [28, 31, 34, 41, 44, 45, 47, 50, 55, 58, 62], *Staphylococcus aureus* –8 [28, 34, 36, 41, 44, 45, 47, 55], *Salmonella spp* –6 [31, 41, 47, 50, 58, 59] –6, *Bacillus cereus* –5 [34, 36, 47, 55, 58], Mesophiles [36, 62], *Enterobacteriaceae* [34, 36], Cholera [37, 51], and Typhoid [37, 51] –2 each, *Clostridium perfringens* [34], *Pseudomonas spp* [55], *Aureobasidium pullulans* [42], *Shigella spp* [58], Diarrhoea [51], and Dysentery [37] –1 each. Second, Fungi: Yeast/Mould –2 [45, 62], *Fusarium oxysporu* [50], *Botrytis cineria* [42], *Alternaria alternate* [42], *Eurotium herbariorum* [42], *Aureobasidium pullulans* [42], *Rhizopus spp.* [50], *Penicillium spp.*, *Fusarium spp.* [50], *and Aspergillus flovlis* [50] –1 each. Third, Virus: Viral hepatitis –1 [37]. Fourth, Drugs: Amoxicillin [53], Chlortetracycline [53], Ciprofloxacin [53], Danofloxacin [53], Doxycycline [53], Norfloxacin [53], Oxytetracycline [53], Sulfadiazine [53], and Tylosine [53] –1 each; and Fifth, Chemicals: Chlorpyrifos –1 [56].

We found groups of ready-to-eat foods with high level of contaminants raising public health concerns. These include, Salad and vegetables –5 [31, 42, 44, 47, 58], sliced mango, meat pie, snail khebab, fried yam, sliced water melon, jollof rice, and roasted plantain –3 [31, 36, 47], pepper sauce –2 [36, 47], macaroni [44], fufu [44], cocoa drink [44], guinea fowl and beef [59], mashed "kenkey" [45], omo-tuo, akple, and rice [36], and sausage and fried fish [47] –1 each.

### Preventive measures against contamination of ready-to-eat food in Ghana

The review further identified some measures required by the stakeholders (regulators, managers, food handlers, and consumers) in attaining a national food safety system in Ghana. First, enforcement –14 [31, 33, 35, 41, 45, 46, 48–51, 54, 56, 58, 63], public education –13 [20, 26, 27, 29, 30, 38, 41, 46, 47, 50, 51, 56, 60], training/ capacity building [8, 35, 39, 40, 48, 57], monitoring and supervision [8, 29, 36, 46, 47, 51], and surveillance/isolation of microbes and other elements [29, 42, 43, 55, 59, 61] –6 each, punishment –4 [39, 45, 46, 50], collaboration –2 [20, 61], others –1 [20]. Second, the need to provide education and training –12 [8, 29, 30, 35, 38, 39, 41, 46, 47, 50, 52, 62], workplace policies and food safety culture –7 [20, 29, 32, 36, 44, 52, 57], provide facilities –6 [8, 29, 33, 48, 49, 55], surveillance/isolation of microbes and other elements –5 [29, 43, 55, 59, 61], medical screening and motivation –2 [33, 47], and others –1 [33]. Third, the need for improved personal/food hygiene among food handlers –13 [8, 28, 34–36, 41, 44–46, 50, 52, 55, 62], medical screening –2 [33, 47], prevention of cross contamination [44], time and temperature control [46], and good sanitation [45] –1 each. Fourth, consumers need to observe personal/food hygiene –8 [28, 34, 35, 41, 44–46, 62], prevention of cross contamination [44], time and temperature control [46], and consumer pressure [36] –1 each.

## Discussion

The purpose of this scoping review is to explore food contamination in Ghana, between 2001–2022.

### Prevalence and regional distribution of studies on food contamination in Ghana

There are more studies on food contamination conducted in the Ashanti, Central, Eastern, and Greater Accra Regions the rest of the regions in Ghana. The concentration of studies around these four regions could possibly be attributable to the high incidence of food contamination recorded. Given that Greater Accra and Ashanti Regions are the two biggest cities in Ghana, and are more densely populated, residents are also likely to patronise more of food-vendor services. Thus, the more outlets that prepare and provide food services to many consumers, the likelihood of recording high contamination if safety of the food is not paramount. Meanwhile, this pattern of prevalence is consistent with Omara et al. [64], where incidence of food contamination was found to be prevalent in some particular regions. The reason for this pattern may be because of the cosmopolitan nature of these regions [19, 65]. Thus, it is essential that food safety is placed high in such regions and their cities to prevent health implications associated with poor compliance to food safety practices.

### Food safety-related risks of public health concern in Ghana

Regulatory bodies tasked to ensure food safety in Ghana have, to a large extent, failed to enforce the regulations, carry out monitoring and supervision, and collaborate with each other in the discharge of their duties. This omission could be because they lack adequate logistics and personnel [66]. Again, managers/supervisors of food service outlets failed to provide adequate infrastructure and facilities, training, monitoring and supervision to their employees, as also observed by previous studies [66–69]. Moreover, food handlers are implicated for poor personal, food-safety and hygiene practices, cooking in unclean environment, poor sanitation, inadequate knowledge and skills, poor time and temperature controls, insufficient formal education and training, and medical screening [15, 18, 66, 70], which could increase food contamination and cause food-borne illness to consumers. Therefore, consumers need to take food

safety practices into their own hands by observing proper personal/food hygiene and safety practices [67].

## Microbial quality of ready-eat-food with high public health risks

The high prevalence of contaminated foods reported in this review is a cause for worry. Most of the contaminants reported can pose serious public health and safety risk to the population [71]. Consumption of foods infected with micro-organisms like *E. coli*/faecal coliforms, *Staphylococcus Aureus*, cholera, and typhoid can result in short-to-long-term absence from work and school, hospitalization, or even death. Other contaminants like *Clostridium Perfringens*, Viral hepatitis, and Chlorpyrifos can impact an entire family or community in a very significant way [72, 73]. This raises public health risk and safety implications of the high incidence of contaminated food.

We also identified and drew attention to some ready-to-eat foods with high levels of contaminants raising public health concerns. Some of these foods include salad, vegetables, sliced mango, meat pie, and snail khebab. These foods are of public health concern because these are popular ready-to-eat foods patronised widely by children, women, men, rich, poor, students, farmers, mechanics, etc.; this coheres with Akparibo et al. [74] who reported similar findings. Unfortunately, weak enforcement, monitoring and supervision on the part of regulators and managers, and inadequate knowledge, poor personal and environmental hygiene, austere economic conditions in Ghana now, and irresponsibility on the part of food vendors and consumers make these foods a clear and present public health threat to the population [6]. Therefore, school and working hours lost to hospitalisation and recuperation could further increase the vulnerabilities of the public [18, 75].

## Preventive measures against food contamination

It is important that food safety regulations are enforced, public education, training/capacity building, monitoring and supervision, surveillance/test for microbial and other elements are increased. Moreover, collaboration, and punishing offenders should be important for reducing the level of food contamination in ready-to-eat food in the country [16, 17, 66]. Similarly, education and training of food handlers, ensuring adequate supervision, providing workplace policies and establishing food-safety culture, provision of adequate facilities, surveillance/testing for microbial agents in food, medical screening, and motivation are critical measures to promote food safety [16, 17]. Therefore, effective protection from the public health risks associated with contaminated foods can be attained through collaborative effort [76]. That is, the regulators and managers, especially, must commit to their mandate and be resolved to protecting consumers from food-related public health and safety risks [77, 78].

## Limitations

Though this review is first to comprehensively explore food contamination, prevalence and preventive measures, it also has some limitations. First, the review was wholly based on published and grey literature, hence may contain biases contained in those findings and conclusions. Second, we restricted the review to only articles published in English, that could affect the outcome of this review. Regardless of these confounding elements, a sample of 40 articles included in this review is sufficient to attain reliability and dependability of the conclusions reached. Moreover, one of the authors is an authority in Food Science, another, an authority in Environmental and Occupational Health and Safety, and another teaches Food Nutrition.

## Conclusions and recommendations

We observed that to stem food safety-related public health risks in Ghana, the key regulatory bodies (FDA, EHS, MLGDR, GTA, VSD, and CPA), must scale-up: i) enforcement, ii) public education, iii) training/capacity building, iv) monitoring and supervision, v) surveillance/ microbial quality test, vi) collaboration, and vii) punishment of food safety offenders. Contaminated food is a public health risks that could result in death, short to long term morbidity, loss of funds, and threatens to displace Ghana's efforts at achieving the SDG (Specifically, SDG 2- to eliminate hunger, attain food security, enhance nutrition and stimulate sustainable agriculture). The regulatory bodies must recommit to their mandate to address the status quo.

Though a lot of studies exist on food safety in Ghana, none gave a clear and comprehensive account of the prevalence and regional distribution of studies on food contamination. Moreover, the few reviews that exist failed to account for common food safety-related risks of public health concerns. Therefore, this review makes a significant contribution to knowledge on the subject. To attain a national food-control and safety system and guarantee public health and safety, we articulate the following recommendations: 1) government must resource the regulatory bodies to enhance their operational capacity, 2) regulatory bodies should collaborate in carrying out monitoring and supervision of food vendors, 3) managers of restaurants and other food service outlets must provide adequate facilities to engender food safety culture, 4) further, research on ready-to-eat foods posing the highest public health and safety risk and the populations most at risk of food contamination in Ghana.

## Supporting information

**S1 Checklist. Preferred Reporting Items for Systematic reviews and Meta-Analyses extension for Scoping Reviews (PRISMA-ScR) checklist.**
(DOCX)

## Author Contributions

**Conceptualization:** Nkosi Nkosi Botha, Edward Wilson Ansah, Cynthia Esinam Segbedzi.

**Data curation:** Nkosi Nkosi Botha, Edward Wilson Ansah, Cynthia Esinam Segbedzi, Sarah Darkwa.

**Formal analysis:** Sarah Darkwa.

**Investigation:** Sarah Darkwa.

**Methodology:** Nkosi Nkosi Botha, Edward Wilson Ansah, Cynthia Esinam Segbedzi.

**Supervision:** Edward Wilson Ansah.

**Validation:** Sarah Darkwa.

**Writing – original draft:** Nkosi Nkosi Botha, Cynthia Esinam Segbedzi, Sarah Darkwa.

**Writing – review & editing:** Nkosi Nkosi Botha, Edward Wilson Ansah, Sarah Darkwa.

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
