## [Decision Letter · Decision Letter 0]

22 May 2023

PONE-D-23-02292Public health concerns for food contamination in Ghana: A scoping reviewPLOS ONE

Dear Nkosi Nkosi BOTHA,

Thank you for submitting your manuscript to PLOS ONE. After careful consideration, we feel that it has merit but does not fully meet PLOS ONE’s publication criteria as it currently stands. Therefore, we invite you to submit a revised version of the manuscript that addresses the points raised during the review process.

We look forward to receiving your revised manuscript.

Kind regards,

Siraj Ahmed Ali

Academic Editor

PLOS ONE

Journal Requirements:

Additional Editor Comments:

The topic of the paper is interesting to be worked out especially in Africa. But it needs to be re-checked and work on it as the reviewers commented it. Please take time and come up with corrected version of the manuscript by incorporating the reviewers comments.

Reviewers' comments:

Reviewer's Responses to Questions

**Comments to the Author**

1. Is the manuscript technically sound, and do the data support the conclusions?

Reviewer #1: No

Reviewer #2: Yes

2. Has the statistical analysis been performed appropriately and rigorously? 

Reviewer #1: N/A

Reviewer #2: N/A

3. Have the authors made all data underlying the findings in their manuscript fully available?

Reviewer #1: Yes

Reviewer #2: No

4. Is the manuscript presented in an intelligible fashion and written in standard English?

Reviewer #1: Yes

Reviewer #2: Yes

5. Review Comments to the Author

Reviewer #1: Manuscript: Public health concerns for food contamination in Ghana: A scoping review (PLOS ONE)

This is a manuscript addressing an important issue of public health concern in Ghana. It is inspiring to see the concept of scoping reviews applied to map out information on such sensitive issues such as food contamination.

The following are my comments and suggestions:

Introduction

1. “Nutrition is sturdily and rapidly … short to long term morbidity.” This statement is repeated verbatim in the abstract. I suggest authors paraphrase or recast the statement to reflect same statement.

2. “Yet, there is inadequate research … in Ghana. Therefore, the purpose … verify the phenomenon of food contamination in Ghana, to inform policy and increase research interest in the area.” These statements create the impression that there is insufficient research yet authors want to verify the phenomenon of food contamination in the midst of inadequate research. Again, I don’t think this review will necessarily increase research interest in the area. Authors should revisit the concept of scoping reviews and why it is conducted in order to provide a good justification for the study.

Methods

3. “We synthesized … from 2001 to 2022.” Authors should justify why the period was chosen for the review work.

4. “used them in surfing eight databases … Emerald Insight, Google scholar and Google search.” Referring to Emerald Insight, Google scholar and Google search is misleading. I suggest authors call them by their appropriate names like search engine.

5. “PubMed (235) .. Google search (3,650,000).” It is not clear what authors did with the report. If they were screened, it is not practical to be able to screen 3,650,000 reports from google search and as well as reports from ScienceDirect, ProQuest, Scopus and Google scholar. Authors have also not stated the screening processes they undertook at this stage to arrive at the 1362 articles and finally 40 articles.

Table 1

6. Since the review was conducted between 2001-2022, can authors explain why ‘Gold Coast’ was added as part of the key words? The name Gold Coast was avoided after 1957, so I find it difficult to comprehend.

7. The inclusion criteria are not smart. For instance, published articles on Ghana, grey literature on Ghana, articles must provide details on … and conclusion. Not all articles will even provide all the details in the earlier statement. Will authors exclude those articles. I suggest authors come out clear with the inclusion and exclusion criteria.

Results

8. “Study took … 23 published and 17 grey.” Grey literature has been mentioned at this stage but it is not stated in the methods. Also, their sources as well as the type of grey literature have not been mentioned.

9. “We found that food contamination is more prevalent in Greater Accra” The statement is problematic. It is only food contamination prevalence or estimates that can be used to support such statement, which is not mentioned here. Authors should recast the statement if more studies on food contamination were done in Greater Accra compared to the other regions.

10. It is difficult to follow the results since number of studies are not initially mentioned in the narratives on each theme or research question. I suggest authors state the number of studies that was found on specific theme as well as studies that fall under specific descriptions under the themes. Authors should look at all the themes from this point and recast them again.

11. The authors should properly synthesize the findings of included articles to reflect the issues of food contamination in Ghana.

Discussion

12. The introductory part of the discussion gives a narration of all the findings that are about to be discussed in the section instead of reiterating the purpose of the study in terms of the themes or research questions that was set out for scoping review. Authors should recast those portions of the discussion.

13. Due to the fact that the results are not mapped out well, almost everything in the results are being discussed, which makes it difficult to identify the issues that are sensitive and requires attention. The authors should rewrite the section again.

Table 2

14. The authors should regroup the articles under the research questions instead of leaving them under one umbrella. This is because some specific studies may have addressed specific research questions and lumping all together makes the writing of results difficult as it was experienced in the work.

15. How data was charted was not explained in the methods section. Authors mentioned of themes, which is likely to be headings used for data charting. Authors should give an information of how it was done in the methods.

Figure 1

16. From the review, about eight databases (where some are search engines) were searched, so one will expect to see the number of articles or hits that resulted after those searches, which seems to be missing here. Authors should look at the figure again.

17. Again, additional records through other sources were not mentioned in the methods but it’s here. Authors should relook at it.

18. A narrative should be written for the figure 1 in the methods. It is difficult to understand the figure and what was done, which hinders duplication of results.

19. It is not clear how consultation was done to retrieve articles to be included in the scoping review work.

General comments

20. The idea to use scoping review is good but its concepts are not properly applied in this work. Authors should revisit the concept and follow the procedures to arrive at a more scientific conclusion.

Philip Apraku Tawiah

School of Public Health, KNUST, Kumasi, Ghana.

School of Pharmacy, UHAS, Ho, Ghana.

Reviewer #2: Let me congratulate authors for taking time to research into such an important phenomenon. I find the following comments useful for enriching the content of the manuscript.

Abstract

The results section of the Abstract should be re-written in order to ensure increased clarity to the findings reported under the different categories.

Introduction

“…Yet, there is inadequate research into the public health and safety threats of contaminated foods in Ghana”

How to you substantiate this?

Methods

Search Strategy Applied

i. Was your search restricted to only English language literature?

ii. What were the other inclusion/exclusion criteria used?

iii. Within the search engine/databases, did you make use of limiters?

iv. Were Boalean operators used?

v. Did you use Word truncation?

Was a data extraction tool developed and used? What were the components of this tool? Thus, can you give a description of this tool?

How did you analyse your secondary data? Can you give some descriptions to the data analysis process.

6. PLOS authors have the option to publish the peer review history of their article (what does this mean?). If published, this will include your full peer review and any attached files.

Reviewer #1: **Yes: **Philip Apraku Tawiah

Reviewer #2: No

---

## [Author Response · Author response to Decision Letter 0]

25 May 2023

Reviewer #1

Response:

We very much appreciate the observations made by the reviewer and consider the queries very useful in improving the overall quality of the manuscript. Our specific responses to the queries are as follows: 

Introduction

Query 1: Very good observation. Comment addressed in the manuscript. 

Query 2: Well noted and addressed in the manuscript. 

Query 3: By our exclusion criteria, we considered articles published prior to 2001 as too old for this review and thus excluded them. 

Query 4:

Issue addressed.

Query 5: As regards how the Google search result (3,650,000) was handled, our response is follows:

The data search went through two levels. At level one, we applied words like “food” OR "contamination” OR “risk” OR “Ghana”, which is intended to gauge the volume of data available on the subject. The results from the initial search included: PubMed (235), JSTOR (1,466), ScienceDirect (59,120), ProQuest (44,177), Scopus (59,120), Emeralds Insight (227), Google scholar (91,200), Google search (3,650,000). With additional words (see Table 1) added, a second level search was conducted and returned the following: PubMed (86), JSTOR (94), ScienceDirect (245), ProQuest (211), Scopus (112), Emeralds Insight (82), Google scholar (127), Google search (405). In all, 1,362 articles were located. The search exercise spanned April 21st, 2022, to June 19th, 2022. Accordingly, the manuscript has been recast to reflect this. Moreover, a PRISMA-statement on the screening process submitted to the journal for further guidance.

Table 1

Query 6: Why “Gold Coast?” 

On the use of “Gold Coast” as a key word. We uphold the observations made by the reviewer and accordingly deleted it from the manuscript.

Query 7: Respectfully, the articles went through a rigorous screening process and must meet the inclusion and exclusion criteria defined in Table 1. So, in addition to serials 1 and 2 of the inclusion criteria, the articles, be it grey (mainly masters’ and doctoral dissertations/theses) or published articles must provide details on food safety in Ghana covering 2001 – June, 2022; and must provide details on methodology, sample/population, study area, causes of food contamination, microbial isolation, high risk foods, prevention of food contamination, and conclusion. 

Results

Query 8: 

Regarding the grey literature, we were clear in our intention that grey literature would be part of the review and provided for it in the inclusion criteria in Table 1. Besides, since the results captured details of the number of published and grey literature included, we did not consider it prudent to capture that somewhere again in the methods. Regarding the source and type of grey literature, these were mainly dissertations/theses. Moreover, sources and details of all articles included in this review are provided in Table 2.

Query 9: We fully uphold the observations made by the reviewer and accordingly effected necessary corrections in this portion of the manuscript. Thus, we amended to read “prevalence and regional distribution of studies on food contamination”. That is, “we found that there were far more studies on food contamination in Greater Accra than in any other region in Ghana”. 

Query 10: The results were carefully presented according to the themes and indicated the number of articles that discussed these specific themes. Consider for example the following:

Food Safety-Related Public Health Risks in Ghana

Considering this specific theme and covering the issue of “weak enforcement of regulations” – 3 articles (Amaami et al., 2017; Seidu, 2020; Wuliyeng, 2013), “inadequate monitoring and supervision” – 2 articles (Ekow, 2018; MacArthur, 2007), and “logistical limitations and ineffective collaboration” – 2 articles (Amaami et al., 2017; MacArthur, 2007)… 

Microbial Quality and Foods with High Public Health Risks 

On this and covering “bacteria: E. coli/faecal coliforms” – 11 articles (Abakari et al., 2018; Agyarko, 2021; Annor et al., 2011; Aovare, 2017; Appietu et al., 2020; Bansah, 2018; Feglo et al., 2012; Kortei et al., 2020; MacArthur, 2007; Mwini et al., 2019; Owusu, 2015), “Staphylococcus aureus” – 8 articles (Agyarko, 2021; Annor et al., 2011; Aovare, 2017; Appietu et al., 2020; Feglo et al., 2012; Kortei et al., 2020; Mensah et al., 2002; Owusu, 2015), …

Preventive Measures against Contamination of Ready-to-Eat Food in Ghana

On this theme and covering “enforcement” – 14 articles (Abakari et al., 2018; Aglidza, 2019; Agyarko, 2021; Amaami et al., 2017; Amedewonu, 2020; Aovare, 2017; Bansah, 2018; Ghartey, 2019; Kasu et al., 2022; Konlan, 2019; MacArthur, 2007; Manko, 2018; Odonkor et al., 2020; Wuliyeng, 2013), “public education” – 13 articles (Ahmed, 2018; Akabanda et al., 2017; Akonor et al., 2013; Amedewonu, 2020; Aovare, 2017; Ayensu, 2020; Ayinpokaapegyine, 2016; Kasu et al., 2022; MacArthur, 2007; Monney et al., 2014; Odonkor et al., 2020; Ovai et al., 2019; Owusu, 2015)…

Thus, the concerns raised by the reviewer have already been sufficiently addressed. 

Query 11: This concern has been adequately addressed and catered for already. Respectfully, there is no action required on this.

Discussion 

Query 12: Observation well noted and accordingly addressed. 

Query 13: Our response to “Query 10” sufficiently addressed this observation. Both the results and discussion were organised according to the themes/objectives of the review. Therefore, no clear action required on this.

Table 2

Query 14: Table 2 broadly provides further and better details on the articles included in the review. Therefore, the table is NOT intended to provide details on themes, rather, to give readers further details on the specific articles included in the review. Moreover, what the reviewer seeks for is sufficiently provided for in the results. 

Query 15: As already provided in response to “Query 7”, the methods is logically presented and explained to ensure ease of replicability. The articles went through adequate distillation and rigorous screening process consistent with the inclusion and exclusion criteria defined in Table 1. So, in addition to just serial 1 and 2 of the inclusion criteria, the articles, be it grey (mainly masters’ and doctoral dissertations/theses) or published, articles must provide details on food safety in Ghana covering 2001 – June, 2022; and further provide details on methodology, sample/population, study area, causes of food contamination, microbial isolation, high risk foods, prevention of food contamination, and conclusion. 

Figure 1:

Queries 16-20: The Figure 1, the PRISMA diagram, was prepared consistent with the standard presentation format defined in the search strategy and PRISMA statement submitted to the journal during the first submission. Therefore, the PRISMA diagram is supposed to compliment the narrative provided under the search strategy and Table 1. On the issue of the “additional records through other sources”, these are essentially articles already in the possession of the authors and others volunteered by friends, and collated for this purpose. The search strategy provided sufficient information on the PRISMA and this is consistent with the design of scoping review, which is a bit different from systematic review. While I see reason in the observations made by the reviewer, a very broad consultation with other scoping reviews published in this very journal was conducted to ensure that the design was consistent with that for scoping review. 

Reviewer #2

We are grateful to the review for making time to review the manuscript, and overall, we find his comments very useful and progressive. Meanwhile, our specific responses to the queries raised are as follows; 

Abstract

We find the comments appropriate and accordingly recast portions of the results.

Introduction 

Very good observation, and we have accordingly recast the purpose to make it fit the concerns raised. 

Methods

i. Yes, and well captured in the search strategy on Table 1. 

ii. This is addressed in the search strategy.

iii, iv, & v. Limiters, Boalean operators, and extraction guideline was developed and used during the data search. The specific words applied, the levels of search, and the number of data returned from the search provided in the search strategy, including Table 1 is a part.

---

## [Editor Report · Decision Letter 1]

2 Jul 2023

Public health concerns for food contamination in Ghana: A scoping review

PONE-D-23-02292R1

Dear Dr. BOTHA,

We’re pleased to inform you that your manuscript has been judged scientifically suitable for publication and will be formally accepted for publication once it meets all outstanding technical requirements.

Kind regards,

Siraj Ahmed Ali

Academic Editor

PLOS ONE
---

## [Editor Report · Acceptance letter]

3 Aug 2023

PONE-D-23-02292R1 

Public health concerns for food contamination in Ghana: A scoping review 

Dear Dr. Botha:

I'm pleased to inform you that your manuscript has been deemed suitable for publication in PLOS ONE. Congratulations! Your manuscript is now with our production department. 

Kind regards, 

on behalf of

Dr. Siraj Ahmed Ali 

Academic Editor

PLOS ONE